# *Russula orientalovirescens* sp. nov., a common Southeast Asian edible fungus is different from the European look-alike *R. virescens*

Komsit Wisitrassameewong[1,2]*, Slavomír Adamčík[3,4], Katarína Adamčíková[3,5], Song-Ming Tang[6], Narumon Tangthirasunun[7], Boontiya Chuankid[8], Olivier Raspé[8,9,10]*

**1** Department of Biotechnology, Faculty of Science, Mahidol University, Bangkok, Thailand, **2** National Biobank of Thailand, National Science and Technology Development (NSTDA), Pathum Thani, Thailand, **3** Laboratory of Molecular Ecology and Mycology, Institute of Botany, Plant Science and Biodiversity Center, Slovak Academy of Sciences, Bratislava, Slovakia, **4** Department of Botany, Faculty of Natural Sciences, Comenius University in Bratislava, Bratislava, Slovakia, **5** Department of Plant Pathology and Mycology, Institute of Forest Ecology, Slovak Academy of Sciences Zvolen, Nitra, Slovakia, **6** College of Agriculture and Biological Science, Dali University, Dali, Yunnan, China, **7** Department of Biology, School of Science, King Mongkut's Institute of Technology Ladkrabang, Bangkok, Thailand, **8** School of Science, Mae Fah Luang University, Chiang Rai, Thailand, **9** Meise Botanic Garden, Meise, Belgium, **10** Service Général de l'Enseignement Supérieur et de la Recherche Scientifique, Fédération Wallonie-Bruxelles, Brussels, Belgium

* komsit.wis@mahidol.ac.th (KW); olivier.raspe@botanicalgardenmeise.be (OR).

## Abstract

Green-cracking Russulas are edible fungi that are widely consumed and traded in Southeast Asia. Asian collections of this morphotype were frequently identified as *R. virescens* in local literature. Multilocus phylogenetic analyses of ITS nrDNA, *rpb*2 and *tef*1 regions presented in this study strongly supported that the majority of green cracking *Russula* collections from Southeast Asia represent a species different from European *R. virescens* and these collections are described here as *R. orientalovirescens* sp. nova. Analysis of ITS barcoding region confirmed that published sequence data from China, Laos and Myanmar reported this species as *R. virescens.* In addition, this analysis showed that the species is widely distributed in Southeast Asia from Malayan Peninsula to Japan, preferring areas with dry season, and is associated with coniferous and deciduous trees as well as heterotrophic plants. Morphological analyses and detailed comparison with recent collections of *R. virescens* showed that *R. orientalovirescens* differs from the latter by larger spores and shorter and more abundant pileocystidia. Green-cracking *Russula* species with distinctly areolate pileus formed a monophyletic lineage where our new species is grouped with Asian *R. viridirubrolimbata*, European *R. virescens* and North American *R. parvovirescens*. Few publicly available ITS sequences from Southeast Asia clustered with either European or North American species suggesting that the phylogenetic lineage of green-cracking Russulas urgently require further attention.

**Data availability statement:** All sequences used in the phylogenetic analyses are available from GenBank (https://www.ncbi.nlm.nih.gov/nucleotide/; see Table 1). The final alignments, as well as raw micromorphological data are available from Zenodo (https://doi.org/10.5281/zenodo.14973678).

**Funding:** O. Raspé: Mae Fah Luang University grant 641A01003, "Survey of edible fungi in dry dipterocarp forests of Chiang Mai Province, Thailand Komsit Wisitrassameewong: the Ecological Monitoring and Plant Specimen and Barcode References project P2250745, National Science and Technology Development Agency Slavomír Adamčík, Katarína Adamčíková: the Slovak Research and Development Agency projects APVV-15-0210 and APVV-20-0257.

**Competing interests:** The authors declare no competing interests.

## Introduction

*Russula* is a genus of agarics (fungi, Agaricomycotina, Russulales) of high ecological importance because its members are very diverse ectomycorrhizal root symbionts of trees and dominate several types of ectotrophic forest soil ecosystems [1]. Among hundreds of described species, there are only few species collected for food and traded, and this economic activity is especially popular in Southeast Asia [2–4]. Among the most common edible and traded fungi of Thailand are green-cracking Russulas which are recognized and named as *R. virescens* (Schaeff.) Fr. in local scientific and professional literature [5–8]. *Russula virescens* has been known as an edible fungus for centuries all over Europe and has various popular names in different languages and dialects [9,10]. Its popularity is bolstered by easy identification, it is the only green areolate *Russula* in the European subcontinent [11,12]. The species originally described from Europe was reported also from other continents, specifically from the United States [13–17], Central America [18] and Southeast Asia [8].

*Russula virescens* is the type species of *Russula* subsection *Virescentinae* Singer, the group defined by usually areolate pileus surface and typical "virescens-type" structure of pileipellis formed by longer subulate terminal cells originating from one or multiple shorter ellipsoid or globular elements forming almost an epithelium [11]. However, recent phylogenies suggest that areolate pileus is not a diagnostic character and the phylogenetic lineage of *R. virescens* includes also *R. mustelina* Fr., a species without areolate pileus [19]. This group is classified in *R.* subgenus *Heterophyllidiae* Romagn. and bears some typical characters of this subgenus like the absence of suprahilar amyloid spot on spores, and one-celled pileocystidia with weak reaction to sulfovanillin and a single apical knob [20].

Soon it became apparent that at least part of so-called *R. virescens* in areas distant from Europe represent different species, and *R. parvovirescens* Buyck, D. Mitch. & Parrent was described as the first look-alike species from the United States [21]. Among the seventeen *Virescentinae* species reported from China, India, Japan, Pakistan, Papua New Guinea and Thailand [22–33], only four have similar colour to *R. virescens*. These green Southeast Asian species, *R. prasina* G.J. Li & R.L. Zhao [29], *R. xanthovirens* Y. Song & L.H. Qiu [32], *R. aureoviridis* Jing W. Li & L.H. Qiu [24] and *R. sribuabanensis* Paloi, Suwannarach N. & Kumla J. [31] have a cuticle that is not distinctly areolate and is only cracking near the pileus margin. All collections of green distinctly areolate Russulas from Southeast Asia have so far been identified as *R. virescens* or as *R.* cf. *virescens*, even when using in-depth and specifically focused phylogenetic analyses [34]. There still recent Asian records of *R. virescens* provided with molecular data, e.g., from China [35], Lao [36], Myanmar [37], and Sri Lanka [38]. However, these publications used sampling covering limited geographical areas and are based mainly on ribosomal DNA regions. ITS sequences of Thai specimens in public sequence databases are often identified as *R. virescens*, but Kaewkrajang et al. [39] reported morphological differences between Thai and European green-cracking Russulas.

Green-cracking Russulas are not only important in ethnomycology of Southeast Asia, but they are also source of interesting bioactive compounds useful in medicine that were specifically studied on material from this geographical area [40]. However,

some *Virescentinae* are also known to have unpleasant odours that may be potentially harmful when used as food, for example *R. chlorinosma* Burl. has persistent intensive odour of chlorine [41]. Because of this, it is particularly important to recognize correctly morphological and phylogenetic species concept within *Virescentinae*. Studies using multilocus phylogenetic analyses repeatedly confirmed that majority of *Russula* species in Southeast Asia are endemic to this area, albeit similar to European species [42]. Based on the molecular and morphological differences previously reported between the Asian green-areolate collections referred to as *Russula virescens* or *R.* cf. *virescens* in the literature on the one hand, and other green Virescentinae species and the European *R. virescens* on the other hand, we hypothesized that there is at least one additional Asian species of green-cracking *Russula*. Our aim was to confirm if there is phylogenetic signal (using both ITS and multi-locus data) to distinguish our recent Thai collections from European *R. virescens* and other described similar Asian *Virescentinae* members, and also to identify morphological differences between them in case they represent a distinct species.

## Materials and methods

### Studied collections and morphological observations

This study is based on five Thai collections of green-cracking Russulas (OR1607, OR1619, OR1623, OR1687 and OR1717) collected in community forests in Chiang Mai and Chiang Rai provinces. Collecting complied with the Thai Plant Varieties Protection Act of 2542 (1999). The collecting sites were represented by dry, semi-deciduous, Dipterocarpaceae-dominated forests with some Fagaceae admixture, between approximately 700 and 1050 m a.s.l. For the purpose of the study, we also included new molecular data of five European collections of *R. virescens*, four of which were from Slovakia and one from Belgium, two Slovak collections of *R. mustelina* Fr. and an additional collection of green-cracking *Russula* identified as *R. viridirubrolimbata* from China. The specimens collected in Thailand were dehydrated at 45°C and deposited in the herbarium of Mae Fah Luang University, Thailand (MFLU). Slovak collections are deposited in SAV herbarium. Macroscopic characters were observed on fresh material. Size, colour and features of pileus, lamellae, stipe and spore deposit were recorded. Basidiome colours were recorded in daylight. We used the terminology for macroscopic features described and illustrated in [43]. Spore print colours were described according to [11].

We used dried materials to study microscopic characters. All samples were prepared and observed under an Olympus CX43 (Tokyo, Japan) or a Nikon Eclipse Ni (Tokyo, Japan). For the description of microscopic characters, we followed the measurement template and terminology proposed by [44]. The microscopic structures were rehydrated using aqueous 5% KOH solution and observed in Congo Red [45]. Spore ornamentation is described and illustrated as observed in Melzer's reagent. Line drawings were made from pictures taken using NIS Elements Software (Tokyo, Japan) or prepared using Olympus U-DA drawing attachment at a projection scale of 2000×. Spores were measured in side view in Melzer's reagent, excluding the ornamentation, and measurements are given as $\{(MIN) [AV-SD]–AV–[AV+SD] (MAX)\}_{length} \times \{(MIN) [AV-SD]–AV–[AV+SD] (MAX)\}_{width}$ in which AV = mean value for the measured collection and SD = standard deviation. Q corresponds to spore "length/width ratio" and is given as (MINQa) Qa–Qb (MAXQb), where Qa and Qb are the lowest and the highest mean ratio for the measured specimens, respectively. Lamellae and pileal tissues were treated with sulfovanillin solution [46] to observe the colour change of cystidium contents. The colour change and incrustation of pileal tissues were checked by mounting in Cresyl blue [47] and carbolfuchsin [11]. The statistics of all measurement were taken based on 30 measurements per collection and three collections per species were included in each species description.

### DNA extraction, PCR amplification and sequencing

Genomic DNA of Asian collections was extracted from fresh tissues preserved in 2% cetyl trimethylammonium bromide (CTAB) buffer, or from dried tissues, using a CTAB protocol adapted from [48]. DNA of Slovak collections was extracted from dried basidiomata using E.Z.N.A. Fungal DNA extraction kit (Omega Bio-Tek, Norcross, Georgia) following the manufacturer's instructions. Three gene regions were used to infer phylogenetic relationships and species delimitation of

studied samples: the internal transcribed spacer region (ITS) of the nuclear ribosomal DNA, the second largest subunit of RNA polymerase II (*rpb*2) and translation elongation factor1 alpha (*tef*1). The three regions were amplified using primers ITS1-F/ITS4 [49] for ITS, bRPB2-6F and bRPB2-7.1R [50] or alternatively reverse primers 7cRruss1/7cRruss2 were used [19] for *rpb*2, and EF1-983F/EF1-2218R sometimes combined with internal primers EF1-1567R and EF1-1577F [51], or EF1-983F/EF1-1567R-R (5'-GACHGTRCCRATACCACCRATYT-3'; this study) for *tef*1. The polymerase chain reaction (PCR) mix consisted of 2 ng/µL of template DNA, forward and reverse primers (10 µM), 5x HOT FIREPol® Blend Master Mix (Solis BioDyne, Estonia) and molecular grade water to bring the total volume to 20 µL. All PCR cycling conditions started with initial denaturation step at 95 °C for 15 min. For ITS was followed by 35 cycles each comprising of denaturation at 95 °C for 30 s, annealing at 56 °C for 30 s and elongation at 72 °C for 90 s., a final extension was carried out at 72 °C for 10 min. Cycling conditions for *rpb*2 were the same as described in [50] and for *tef*1 touchdown PCR procedure as described in [52] and [51].

All PCR products were checked using 1% agarose gel electrophoresis. DNA sequencing of the successful PCR products using the PCR primers were performed on an ABI 3700 automated DNA sequencer by U2Bio Thailand for Asian collections and SEQme sequencing company (Dobříš, Czech Republic) for European collections. The resulting raw sequences were manually checked, edited and assembled in Geneious R8 or R10 (Biomatters).

**Phylogenetic analyses**

Contigs and sequences retrieved from GenBank (Table 1) were gathered in MEGA11 [53] or BioEdit ver. 7.2.5 [54] and aligned using MAFFT 7.402 [55]. Partial regions of the small subunit (SSU) and large subunit (LSU) of the nuclear ribosomal DNA were excluded from the ITS region sequences. The ITS alignment was processed with TrimAl [56], using the automated algorithm, to eliminate poorly aligned positions. *Russula floriformis* subsp. *floriformis* and *R. floriformis* subsp. *symphoniae* (subsect. *Substriatinae*) were selected as the outgroup [57]. Prior to multi-locus analyses, single-gene tree topologies were compared and examined for conflicts at nodes with bootstrap support value (BS) above 70%. A combined dataset of ITS-*rpb*2-*tef*1 was assembled in BioEdit 7.2.5 [54] and partitioned Maximum Likelihood (ML) and Bayesian Inference (BI) analyses were performed. The partition comprised four character sets for the ML analysis: the ITS region, *rpb*2 exons, *tef*1 exons, and introns from *rpb*2 and *tef*1 combined. For the BI analysis, the 5.8S gene was separated from ITS1 and 2 as a fifth character set. Partitioned ML analysis was performed in RAxML 8.2.12, with the GTRGAMMA+I model and the rapid bootstrapping algorithm for 1,000 replicates [58]. The best-fit sequence evolution model was inferred for each character set using the corrected Akaike criterion in jModelest [59]. The following models were obtained: HKY + G for ITS1 + ITS2, K80 for 5.8S, SYM + G for *rpb*2 exons and for *tef*1 exons, and HKY + I for introns. Partitioned BI analysis was executed in MrBayes 3.2.7a [60], using two runs with four chains for 2 million generations, which were sampled every 250th. Before obtaining the consensus tree from the samples, the convergence and burn-in proportion were checked using Tracer 1.6 [61]. All phylogenetic analyses were done using the CIPRES Science Gateway [62].

A second, ITS-only analysis was performed with sequences gathered from public databases to gather more information about distribution of *Virescentinae* in Southeast Asia and for sequences of taxa closely related to the studied species. ITS sequences with at least 3% similarity to sequences of green-cracking Russulas obtained in this study were retrieved from UNITE database [63]. We also included sequences of known species of *R.* subsect. *Virescentinae* from the literature [22–26,28–33]. All sequences were gathered in MEGA11 and aligned using MAFFT 7.402 [55]. The final ITS alignment was analysed under similar conditions as the multi-locus dataset.

NomenclatureThe electronic version of this article in Portable Document Format (PDF) in a work with an ISSN or ISBN will represent a published work according to the International Code of Nomenclature for algae, fungi, and plants, and hence the new names contained in the electronic publication of a PLOS ONE article are effectively published under that Code from the electronic edition alone, so there is no longer any need to provide printed copies.

**Table 1. Sequences used for the multi-locus analyses. Newly generated sequences are indicated in bold. Type specimens are designated with (T).**

| Taxa | Voucher | Country | Genbank accession number | | |
|---|---|---|---|---|---|
| | | | *rpb*2 | ITS | *tef*1 |
| Outgroup | | | | | |
| *R. floriformis* subsp. *floriformis* | SAVF-20573 (T) | Columbia | MT021752 | MT039866 | MT024552 |
| *R. floriformis* subsp. *symphoniae* | SAVF-20574 (T) | Panama | MT021753 | MT039867 | MT024553 |
| Closely related groups in subg. *Heterophyllidiae* | | | | | |
| *R. amoena* | SAFV-3147 | | MT417202 | MT017544 | MT417211 |
| *R. bella* | SFC20170731−02 | South Korea | MT199645 | MT017556 | MT199658 |
| *R. orientipurpurea* | SFC20170725−37 (T) | South Korea | MT199639 | MT017548 | MT199652 |
| *R. phloginea* | CNX530524068 (T) | China | – | MK860701 | MK894877 |
| *R. subbubalina* | RITF4710 (T) | China | – | MW646978 | MW650847 |
| *R. subpallidorosea* | RITF4083 | China | – | MK860697 | MK894875 |
| *R. vesca* | RITF5038 | China | – | MW646984 | MW650851 |
| Subsect. *Virescentinae* | | | | | |
| *R.* aff. *virescens* | 721BB-09–021 | New Caledonia | KU237868 | – | KU238009 |
| *R. albidogrisea* | K15091234 (T) | China | – | KY767807 | MN617847 |
| *R. albolutea* | RITF2653 (T) | China | MW411340 | MT672478 | – |
| *R. albolutea* | RITF4460 | China | MW411341 | – | – |
| *R. albolutea* | RITF4461 | China | MW411342 | – | – |
| *R. aureoviridis* | RITF4709 | China | MW646980 | MW646980 | MW650849 |
| *R. aureoviridis* | H16082612 (T) | China | – | KY767809 | MN617846 |
| *R. luofuensis* | RITF4706 | China | – | MW646973 | MW650842 |
| *R. luofuensis* | RITF4707 | China | – | MW646974 | MW650843 |
| *R. luofuensis* | RITF4708 (T) | China | – | MW646975 | MW650844 |
| **R. mustelina** | **SAVF-2593** | **Slovakia** | **PP736577** | **PP724717** | **PP736583** |
| **R. mustelina** | **SAVF-3124** | **Slovakia** | **PP736578** | **PP724718** | **PP736584** |
| **R. orientalovirescens sp. nov.** | **OR1607** | **Thailand** | **–** | **PQ881605** | **–** |
| **R. orientalovirescens sp. nov.** | **OR1619** | **Thailand** | **–** | **PQ881606** | **–** |
| **R. orientalovirescens sp. nov.** | **OR1623** | **Thailand** | **–** | **PQ308639** | **–** |
| **R. orientalovirescens sp. nov.** | **OR1687 (T)** | **Thailand** | **PQ310350** | **PQ308640** | **PQ310348** |
| *R. pallidula* | RITF2613 (T) | China | MH091698 | MH027958 | MW650852 |
| *R. pallidula* | RITF3331 | China | MH091699 | MH027959 | MW650853 |
| *R. prolifica* | 18BB-06–161 | Madagascar | KU237741 | – | KU237890 |
| *R. pseudopunicea* | BJTC-C335 | China | OP156851 | MW554144 | – |
| *R. pseudopunicea* | BJTC-ZH1389 | China | OP156852 | OP133163 | – |
| *R. pseudopunicea* | BJTC-ZH1392 (T) | China | OP156853 | OP133164 | – |
| *R. subpunicea* | RITF1435 | China | MW411346 | MN833637 | – |
| *R. subpunicea* | RITF2648 | China | MW411345 | MN833638 | – |
| *R. subpunicea* | RITF3715 (T) | China | MW411344 | MN833635 | – |
| **R. viridirubrolimbata** | **HBAU15020** | **China** | **PQ310351** | MT337527 | **–** |
| **R. aff. viridirubrolimbata** | **OR1717** | **Thailand** | **PQ310352** | **PQ308640** | **PQ310349** |
| **R. virescens** | **VHB38** | **Belgium** | **–** | **PQ308642** | **–** |
| **R. virescens** | **SAV F-3178** | **Slovakia** | **PP736575** | **PP724713** | **PP736579** |
| **R. virescens** | **SAV F-4207** | **Slovakia** | **–** | **PP724714** | **PP736580** |
| **R. virescens** | **SAV F-3326** | **Slovakia** | **–** | **PP724715** | **PP736581** |
| **R. virescens** | **SAV F-21192** | **Slovakia** | **PP736576** | **PP724716** | **PP736582** |

 

In addition, new names contained in this work have been submitted to MycoBank from where they will be made available to the Global Names Index. The unique MycoBank number can be resolved and the associated information viewed through any standard web browser by appending the MycoBank number contained in this publication to the prefix http://www.mycobank.org/MB/. The online version of this work is archived and available from the following digital repositories: LOCKSS, PubMed Central.

## Results

### Phylogenetic analyses

The final multilocus alignment of ITS, *rpb*2 and *tef*1 regions consisted of 42 samples of *Russula* subgenus *Heterophillidiae* and was 1,955 bp long, including gaps, and had 799 distinct alignment patterns. All included green-cracking *Russula* collections with typical appearance of *R. virescens* originated from Asia and Europe and were placed in a single strongly supported clade (MLBS = 100%; Fig 1). European collections of *R. virescens* formed a strongly supported (MLBS = 100%) clade sister to all Asian collections of green-cracking Russulas. Asian collections were placed in two strongly supported subclades. The first Asian clade (MLBS = 98%) contained three collections from Southeast China identified as *R. viridirubrolimbata*: HKAS122576 published in [35] and HBAU15011 and HBAU15020 published in [25]. Our collections of green-cracking *Russula* OR1717 from Thailand is placed in this clade on a longer branch and might represent a distinct, closely related species. Our three other green-cracking Russulas from Thailand were placed in the second Asian clade (MLBS = 96%) and are described as *R. orientalovirescens* sp. nov. in this study.

As the result of UNITE search at 3% similarity threshold, 84 collections from South East Asia were retrieved. Additional 12 sequences from USA and 12 sequences from Europe and two sequences or *R. virescens* supposedly collected in Europe with high similarity to *R. virescens* were included in the ITS analysis. General topology of ITS tree was congruent with multilocus tree (Fig 2), green cracking *Russula* species formed a supported clade within *Virescentinae* (MLBS = 89%). Five species clusters and a singleton sequence were supported within this clade. At least one sequence of Asian origin is placed in each of these clades. The highest number of retrieved samples were placed in the clade corresponding to *R. orientalovirescens* (MLBS = 76%). Among 72 samples of this clade retrieved from UNITE, 59 were from China, 8 from Laos, two from Japan and one from each Myanmar, South Korea and Thailand. Only three collections can be linked to symbiotic host plant by sequences originated from roots, OL475404 from *Picea asperata* (China), AB629011 from *Pyrola japonica* (Japan) and LC315886 from Fagaceae (Japan). Four UNITE Chinese sequences and one from Laos were placed in clade of *R. viridirubrolimbata*, together with one of our sequence from Thailand (OR1717). Two sequences from Indonesia from environmental samples associated with *Shorea* formed a distinct clade and likely represent a so far undescribed species. The singleton sequence MW374153 from China possibly represent also additional species in this species complex. Two retrieved Thai sequences and one from Sri Lanka were placed in *R. virescens* clade together with mainly sequences of European origin. One Thai sequence retrieved from UNITE is placed in the clade which probably corresponds to North American *R. parvovirescens*.

### Taxonomy

***Russula orientalovirescens*** sp. nov. Wisitr. & Raspé [urn:lsid:mycobank.org:names:857481] (Figs 3–6).

   **Mycobank number:** 857481

   **Etymology:** 'oriental' means eastern and 'virescens' refers to the European *R. virescens*. The name epithet thus refers to the resemblance to *R. virescens* (by the green cracked pileipellis) and the species distribution in East and Southeast Asia.

   **Diagnosis:** *Russula orientalovirescens* can be distinguished from similar species in subsect. *Virescentinae* by the following combination of characters: pileus surface with areolate cuticle on white background, cuticle at centre varying

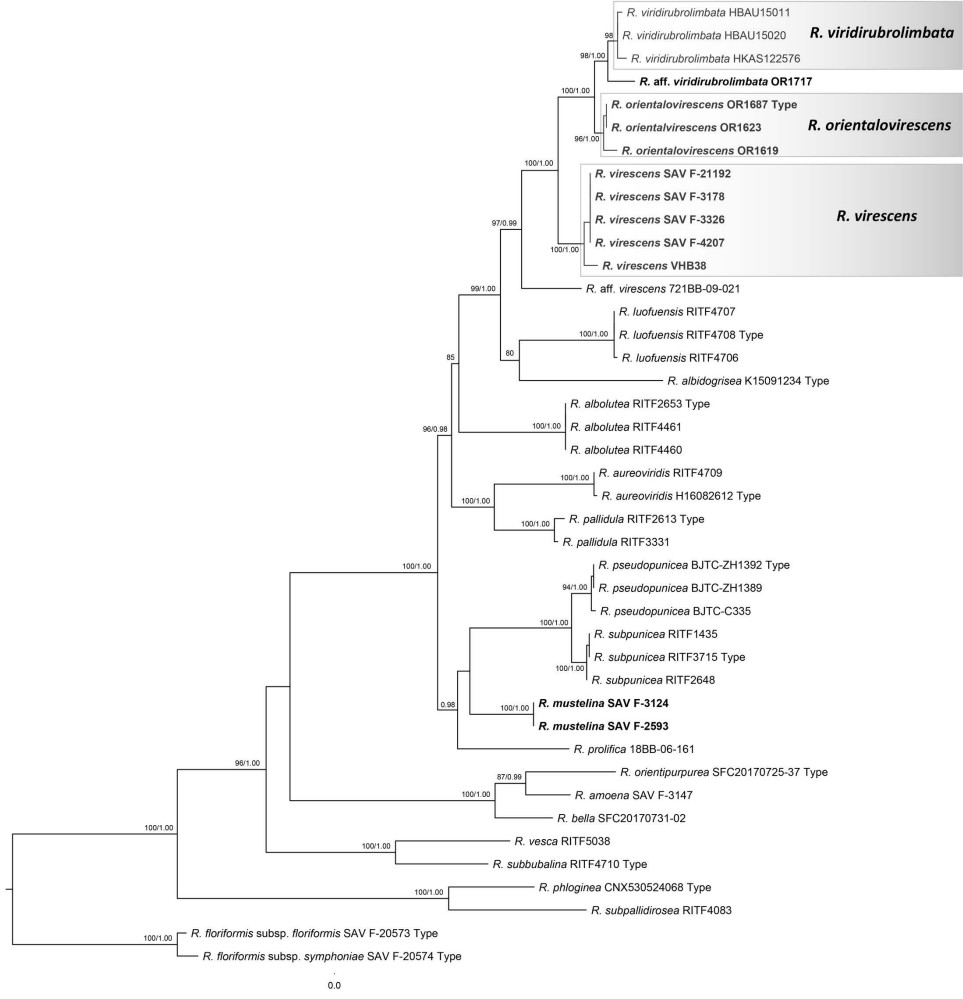

**Fig 1. Maximum Likelihood (ML) phylogeny based on internal transcribed spacer (ITS), the second largest subunit of RNA polymerase II (*rpb2*) and the translation elongation factor 1-alpha (*tef1*) sequences of *R.* subsect.** *Virescentinae* and closely related species. Species in boldface are generated in this study. ML bootstrap values >70 and Bayesian Inference posterior probability >0.9 are shown. Asterisks indicate clades with 100 ML bootstrap values and 1.0 Bayesian posterior probabilities.

from pale yellow to pale green to green, cuticle near margin green to yellowish green; spores broadly ellipsoid to ellipsoid, warts frequently connected into short ridges, low isolated warts numerous; pileocystidia abundant, one-celled, relatively small; terminal cells short to long subcylindrical to subclavate, arising from ellipsoid to globose subterminal cells.

**Holotype:** THAILAND. Chiang Mai Province, Mae On District, Huay Kaew community forest, N18.870°-E99.292°, elev. 700 m, 6 September 2020, O. Raspé 1687 (MFLU-HT21-0044).

**Pileus** medium-sized, 53–87 mm diam., first hemispherical when young, later convex to plano-convex, and with a depressed centre; surface dry, often with whitish pruinose; cuticle continuous and smooth when young, becoming strongly areolate when mature, especially ca. half radius towards margin; margin deflexed, striate; areolae in centre with pale yellow to pale green to green colours, areolae near the margin pale green to green to yellowish green, on paler white background. **Lamellae** white to yellowish white in face view, pale yellowish orange ("blanched almond") in tangential view, close, adnate-emarginate; lamellulae absent; furcations very frequent and near the stipe; edge even, concolorous. **Stipe**

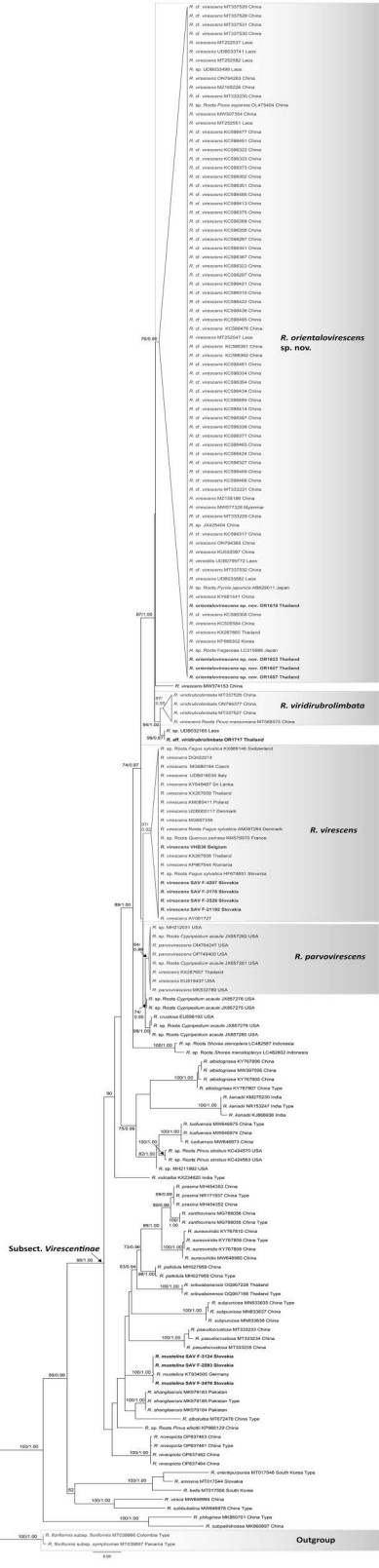

**Fig 2. Maximum Likelihood (ML) phylogeny based on internal transcribed spacer (ITS) sequences of most *R.* subsect.** *Virescentinae* and closely related species. Species in boldface are generated in this study. ML bootstrap values >70 and Bayesian Inference posterior probability >0.9 are shown. Asterisks indicate clades with 100 ML bootstrap values and 1.0 Bayesian posterior probabilities.

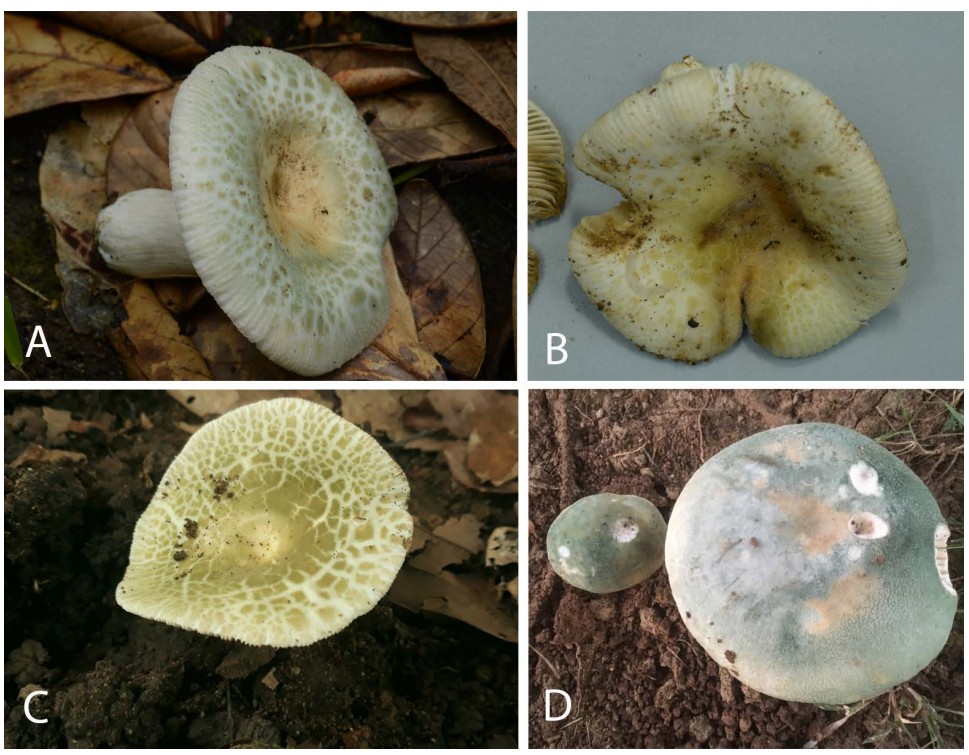

**Fig 3. Basidiomata of described species.** A–C: *R. orientalovirescens* (A: OR1607, B: OR1619, C: OR1687), D: *R. virescens* (SAV F-21192). Photographs: A–C by O. Raspé, D by S. Adamčík.

42–60 × 10–19 mm, subcylindrical to cylindrical, often tapering near base; surface dry, longitudinally striate, white, turning pale brown when bruised. **Context** white, unchanging, compact to firm. **Odour** not distinctive. **Taste** mild. **Spore print** white (1B).

 **Spores** (n = 90) (7.1–)7.4–8.1–8.7(–10.5) × (5.6–)6.0–6.4–6.9(–8.5) μm, broadly ellipsoid to ellipsoid, rarely narrowly ellipsoid, Q = (1.10–)1.19–1.26–1.32(–1.54); ornamentation of moderately distant [(3–)4–6(–7) in a 3 μm diam. circle], amyloid low warts, (0.3–)0.4–0.6(–0.8) μm high, occasionally fused into short chains [(0–)1–2(–3) in a 3 μm diam. circle], short ridges numerous [(0–)1–3(–5) in a 3 μm diam. circle], warts frequently interconnected by lower lines forming an incomplete reticulum, isolated warts common; suprahilar spot moderately large, inamyloid. **Basidia** (29.5–)35.5–41.6–47.5(–49.0) × (7.0–)8.5–9.6–10.5(–11.5) μm, narrowly clavate to clavate, 4-spored, partially with granular contents. **Hymenial cystidia** widely dispersed (< 350/mm²), (39.0–)53.5–65.9–80.0(–105.5) × (4.2–)7.4–8.8–10.1(–13.3) μm, narrowly fusiform to fusiform, apically subacute to mucronate, occasionally slightly moniliform, appendage (1.5–)2.0–4.0(–7) μm, contents heteromorphous, partially filled with granular and short needle-like contents, negative in sulfovanillin; near the lamellae edges (27.0–)48.5–61.1–74.0(–106.5) × (5.0–)6.7–8.2–9.8512.0 μm, dispersed, emergent beyond hymenial layer, subcylindrical to narrowly fusiform, apically round to mucronate, appendage up to 2.0–4.0(–6.5) μm, contents heteromorphous, granular, negative in sulfovanillin. **Lamellae edges** consisting of densely arranged marginal cells, basidia,

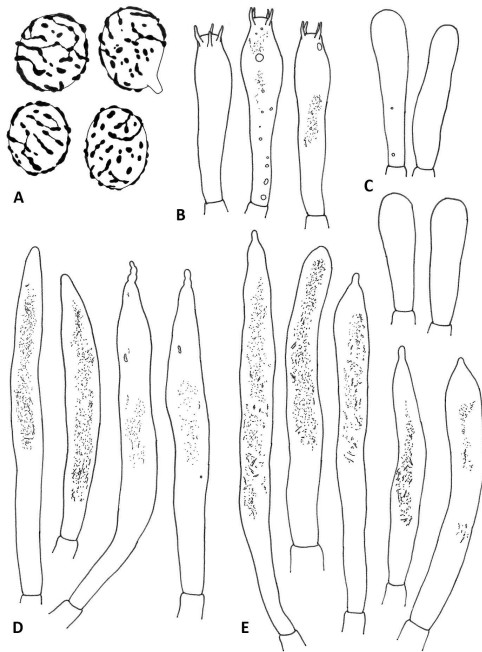

**Fig 4. Hymenial elements of *R. orientalovirescens* (OR1687).** A. Basidiospores, B. Basidia, C. Marginal cells, D. Hymenial cystidia on lamellae side, E. Hymenial cystidia on lamellae edge. Illustration by K. Wisitrassameewong. Scale bars = 10 μm.

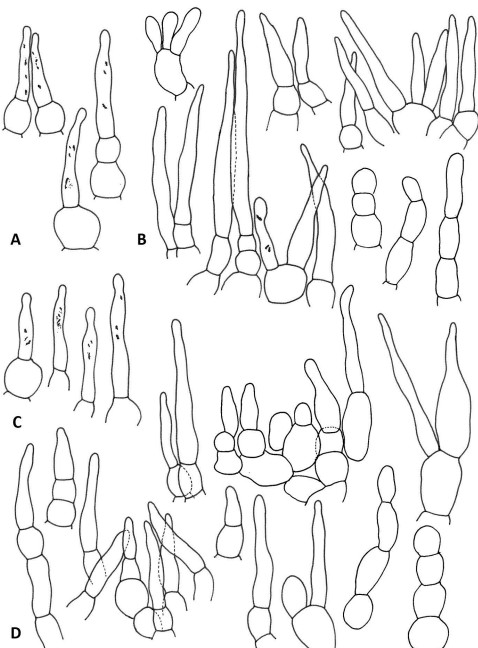

**Fig 5. Elements in pileus cuticle of *R. orientalovirescens* (OR1687).** A. Pileocystidia on pileus centre, B. Hyphal terminations on pileus centre, C. Pileocystidia on pileus margin, D. Hyphal terminations on pileus margin. Illustration by K. Wisitrassameewong. Scale bar = 10 μm.

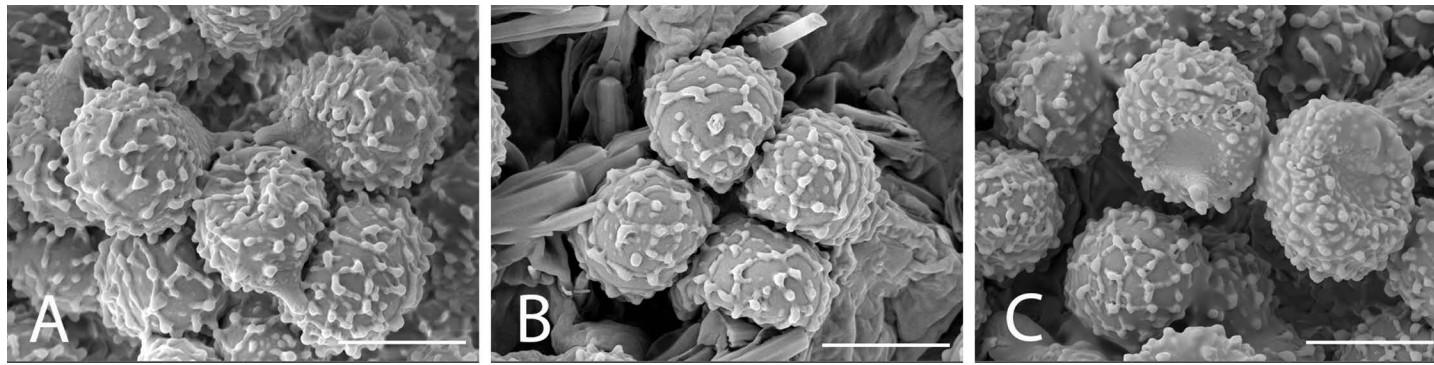

**Fig 6. SEM pictures of *R. orientalovirescens*.** A: OR1607, B: OR1619, C: OR1687. Photographs by B. Chuankid. Scale bars = 5 μm.

and basidioles; marginal cells (15.0–)21.8–<u>25.5</u>–29.5 × (4.6–)5.6–<u>6.7</u>–7.9(–9.1) μm, cylindrical to subclavate, apically round. **Pileipellis** orthochromatic in Cresyl blue, sharply delimited from the underlying context, 75–250 μm deep, two-layered, not gelatinized; suprapellis composed of chains of short cells with attenuated hyphal termination; subpellis composed of layers of ellipsoid to globose cells. **Acid-resistant incrustations** absent. **Hyphal terminations near the pileus margin** ascending or erect, occasionally branched, thin-walled; terminal cells (6.7–)10.9–<u>19.6</u>–28.5(42.5) × (2.7–)4.0–<u>4.9</u>–5.9(–6.8) μm, varying from short and broader to longer and narrower, mainly subcylindrical or broader near base, at times subclavate, apically round, subterminal cells globose to subglobose, arranged in chains, typically unbranched, (5.0–)5.5–<u>8.9</u>–12.0(–19.5) μm diam. **Hyphal terminations near the pileus centre** similar to those near the pileus margin; terminal cells (7.2–)10.2–<u>19.7</u>–29.5(–45.0) × (2.6–)3.6–<u>4.6</u>–5.7(–7.7) μm, mainly subcylindrical, occasionally shorter and subclavate, apically round, subterminal cells globose to subglobose, arranged in chains, typically unbranched, (4.5–)6.0–<u>7.6</u>–9.5(–14) μm diam. **Pileocystidia near the pileus margin** (9.4–)15.0–<u>20.3</u>–25.6(–34.5) × (2.3–)3.2–<u>3.9</u>–4.7(–6.0) μm, small and inconspicuous, somewhat abundant, one-celled, thin-walled, erect from subterminal inflated cells, subcylindrical or tapering towards apex, constricted near apex, apically round, contents partially with crystalline. **Pileocystidia near the pileus centre** (12.0–)16.3–<u>24.1</u>–31.8(–52.0) × (2.3–)3.0–<u>3.8</u>–4.7(–6.1) μm, small and inconspicuous, abundant, similar to those near the pileus margin. Cystidioid and oleipherous hyphae in subpellis and context not observed.

**Additional examined specimens:** THAILAND. Chiang Mai Province, Mae Taeng District, Baan Tha Pha, N19.141°-E98.763°, elev. 1,050 m, 9 August 2019, O. Raspé 1607 (MFLU-HT21-0042). Chiang Rai Province, Mueang District, Doi Pui community forest, N19.817°-E99.866°, elev. 720 m, 20 August 2019, O. Raspé 1619 (MFLU-HT21-0043). —, 26 August 2019, O. Raspé 1623.

**Habit, habitat and distribution:** Usually solitary, in dry, semi-deciduous forests dominated by Dipterocarpaceae (*Dipterocarpus* and *Shorea* spp.) in Northern Thailand. The precise distribution and ecology in neighbouring countries should be further studied.

**Note:** The field appearance of *R. orientalovirescens* is very similar to European *R. virescens* and Chinese *R. viridirubrolimbata*, they all have greenish cracking pileus cuticle. The Chinese species is distinct in having pinkish to reddish colours near pileus margin [25,30,64]. The European species is very similar and we were unable to define any significant macromorphological difference from the new species. However, *Russula orientalovirescens* differs from *R. virescens* in several micromorphological characters, i.e., it has larger spores, wider basidia, larger subterminal cells in pileipellis, and shorter and more numerous pileocystidia near the pileus margin (Table 2). North American *R. parvovirescens* has smaller spores and more inflated (sub)terminal cells compared to the new Asian species (Table 2).
***Russula virescens*** (Schaeff.) Fr. 1836 (Figs 7 and 8).

**Table 2. Comparisons of microscopic characters of *Russula orientalovirescens* vs. *R. virescens* and *R. parvovirescens*. The measurement units of all characters except Q value are in µm. Characters showing distinct differences with *R. orientalovirescens* are highlighted.**

| Characters | *R. virescens* | | | *R. orientalovirescens* | | | *R. parvovirescens* |
| --- | --- | --- | --- | --- | --- | --- | --- |
| | SAV F-3178 | SAV F-3326 | SAV F-21192 | OR1607 | OR1619 | OR1687 | Buyck et al. [21] |
| Spores | 6.4–7.2 × 5.3–5.8 | 7.3–8.2 × 5.5–6.0 | 6.6–7.4 × 5.1–5.7 | 7.5–8.7 × 6.1–6.7 | 7.4–8.2 × 6.0–6.6 | 7.6–9.2 × 6.1–7.2 | 7.5–8.0 × 5.7–6.4 |
| Q value | 1.18–1.27 | 1.29–1.41 | 1.24–1.36 | 1.19–1.34 | 1.18–1.29 | 1.19–1.34 | 1.17–1.26 |
| Basidia | 43.6–51.9 × 8.4–9.2 | 38.0–46.0 × 7.9–9.3 | 52.7–59.4 × 8.4–9.4 | 42.6–49.1 × 9.4–10.8 | 37.3–46.9 × 9.4–10.7 | 32.3–41.3 × 8.2–9.5 | 38–45 × 8–9 |
| Marginal cells | 16.9–27.1 × 5.2–7.6 | 23.4–37.8 × 4.4–6.0 | 22.2–33.5 × 4.9–6.4 | 22.4–29.4 × 5.4–7.2 | 21.1–29.0 × 6.4–8.6 | 22.4–28.7 × 5.8–7.9 | 25–40 × 3–5 |
| Hymenial cystidia on lamellae sides | 74.2–102.0 × 7.2–9.8 | 57.7–80.6 × 7.1–8.5 | 60.2–78.4 × 7.4–10.0 | 68.7–87.5 × 7.4–9.3 | 48.5–67.2 × 7.2–9.6 | 54.4–68.3 × 8.0–10.6 | 60–70 × 8–9 |
| Hymenial cystidia near lamellae edges | 53.8–75.9 × 6.3–7.7 | 38.8–55.0 × 5.3–7.7 | 48.5–66.8 × 5.8–8.1 | 54.3–81.2 × 7.0–9.3 | 39.3–62.4 × 5.1–8.7 | 51.0–68.4 × 8.0–9.4 | N.A. |
| Terminal cells margin | 5.9–22.5 × 4.7–7.5 | 18.1–53.7 × 3.9–8.3 | 11.4–25.9 × 3.9–5.4 | 17.5–33.8 × 3.9–5.8 | 8.6–13.1 × 4.1–5.8 | 13.0–25.6 × 3.8–6.0 | × 6–10* |
| Pileocystidia margin | 30.6–60.7 × 3.2–4.1 | 32.8–56.4 × 2.9–3.8 | 29.7–68.6 × 2.7–3.7 | 16.7–29.7 × 3.0–4.9 | 13.3–24.1 × 3.2–4.4 | 16.2–24.1 × 3.4–4.8 | N.A. |
| Terminal cells centre | 5.7–28.1 × 4.8–8.2 | 16.3–43.7 × 3.3–4.9 | 14.2–31.2 × 4.0–6.3 | 16.8–37.3 × 4.3–6.3 | 11.9–18.4 × 3.3–4.6 | 9.9–16.8 × 3.5–5.1 | × 6–10* |
| Pileocystidia centre | 28.3–68.6 × 3.3–4.2 | 28.0–64.7 × 3.0–4.8 | 23.6–45.3 × 2.3–3.6 | 17.5–37.1 × 3.3–5.5 | 13.6–22.0 × 3.5–4.1 | 20.5–32.0 × 2.7–4.3 | N.A. |

Note:

*The position of the terminal cells (near margin or near center) not differentiated in [21].

**Diagnosis:** Pileus surface with distinct granules or fine areolae towards the margin, dull green to grey or dark green near the centre, context compact to firm, with indistinct and pleasant odour, spore print white; spores broadly ellipsoid to ellipsoid, subreticulate, with small, not amyloid suprahilar spot; hymenial cystidia fusiform or lanceolate, more abundant near the lamellae edges, with heteromorphous contents in almost 3/4 of the volume, weakly react with sulfovanillin; pileipellis of well defined suprapellis of densely arranged globose cells forming almost epithelium and near surface with

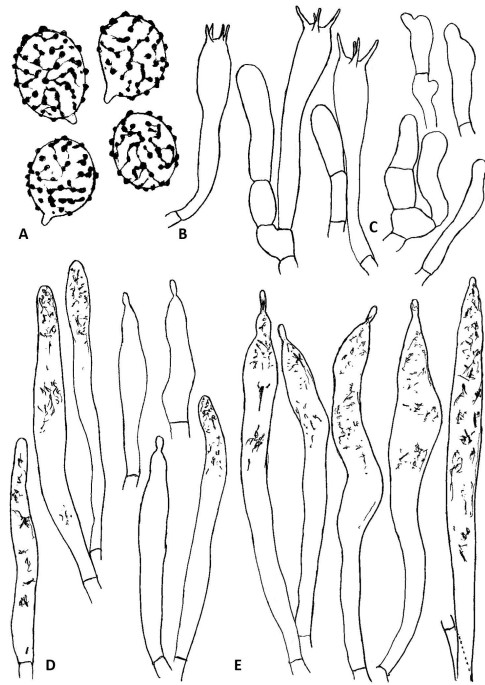

**Fig 7. Hymenial elements of *R. virescens* (SAV-F 3978).** A. Basidiospores, B. Basidia, C. Marginal cells, D. Hymenial cystidia on lamellae edge, E. Hymenial cystidia on lamellae side. Illustration by S. Adamčík. Scale bars = 10 µm.

more elongated loose terminal cells, subpellis strongly gelatinized; pileocystidia inconspicuous, near margin dispersed or absent.

**Pileus** medium-sized to large, 75 mm diam., first hemispherical, later convex to plano-convex, and with a slightly depressed centre, when old usually with deflexed margin; when mature indistinctly striate up to 10 mm; cuticle dry, granulose, soon cracking to small areolae that are max 5 mm large near the margin and gradually pass in ca. half radius to continuous, smooth cuticle near the centre, areolae near the margin dull green or greyish green on cracked placed reveal greyish green gelatinous subpellis, areolae on sun-exposed places discolouring to greyish yellow (2B3) on paler white background, near the centre greenish grey, dull green to dark green and variegated with reddish blonde. **Lamellae** moderately distant, adnate-emarginate, orange white (5A2); lamellulae occasional to rare; furcations very frequent; edge irregular. **Stipe** 80 × 24 mm, cylindrical or clavate, often narrowed near base, longitudinally strongly striate, pitted or rugulose on base white, base soon becoming yellowish white with rusty spots; interior solid. **Context** white, unchanging, compact to firm, but crumbling when cut. **Odour** indistinct, pleasant like *Boletus*, but in dry condition with unpleasant ammonia component. **Spore print** white (1B).

**Spores** (6.0–)6.6–<u>7.2</u>–7.8(–9.0) × (4.7–)5.3–5.<u>7</u>–5.9(–6.2) µm, broadly ellipsoid to ellipsoid, Q=(1.15–)1.22–<u>1.29</u>–1.37(–1.48); ornamentation of relatively small, moderately distant to dense [(3–)5–8(–11) in a 3 µm diam. circle] amyloid warts, (0.2–)0.4–0.6 µm high, usually fused in chains [(0–)1–4(–5) fusions in the circle] and connected by occasional to frequent line connections [(0–)1–4(–5) line connections in the circle], forming subreticulate structure, isolated warts rare or absent; suprahilar spot small, not amyloid, smooth. **Basidia** (36.0–)42.0–<u>48.6</u>–55.5(–60.0) × (6.5–)8.0–<u>8.8</u>–9.5(–10.0) µm, clavate, pedicellate, 4-spored; basidioles first cylindrical, then clavate, ca. 5–8 µm wide. **Hymenial cystidia** widely dispersed, ca. 130–150/mm², (53.0–)62.0–<u>76.9</u>–92.0(–123.0) × 7.0–<u>8.5</u>–9.5(–12.0) µm, fusiform or lanceolate, pedicellate, apically acute, usually with a 2–5 µm long appendage, thin-walled; contents heteromorphous in almost 1/2–3/4

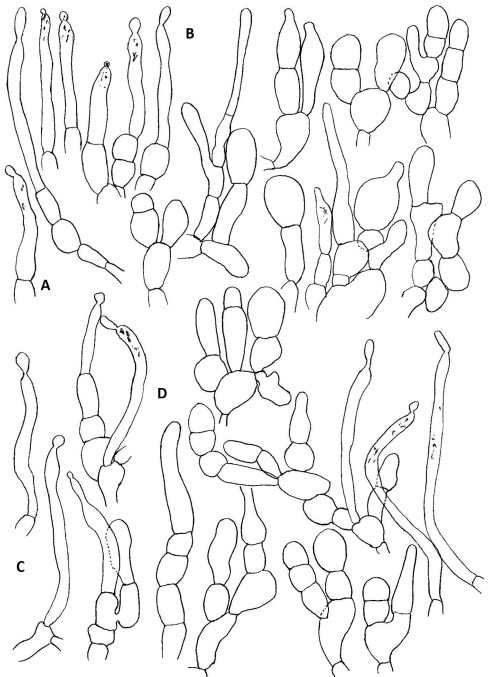

**Fig 8. Elements in pileus cuticle of *R. virescens* (SAV-F 3978).** A. Pileocystidia on pileus centre, B. Hyphal terminations on pileus centre, C. Pileocystidia on pileus margin, D. Hyphal terminations on pileus margin. Illustration by S. Adamčík. Scale bar = 10 μm.

of the volume, granulose-crystalline to banded, apically more dense, almost negative in sulfovanillin; near the lamellae edges more frequent, smaller, (33.0–)44.5–56.4–68.5(–78.0) × (5.0–)5.5–6.8–8.0(–9.0) μm, narrowly fusiform, clavate, lanceolate, subcylindrical, apically acute or obtuse, without or with 2.0–10.0(–13.0) μm long appendage; contents usually optically empty, or granulose only in terminal 1/4 part. **Lamellae edges** sterile; marginal cells (14.0–)20.0–26.8–34.0(–44.0) × (4.0–)5.0–5.7–6.5(–8.0) μm, thin-walled, fusiform, cylindrical, clavate, often flexuous, apically obtuse or constricted, rarely also with small outgrowths or nodulose. **Pileipellis** orthochromatic in Cresyl Blue, not sharply delimited from the underlying context, 250–330 μm thick, strongly gelatinized throughout, well divided in 55–110 μm thick, not gelatinised suprapellis of subglobose, densely arranged cells forming epithelium and near surface of ascending or erect, more loose hyphal terminations; and in a well-defined, 200–280 μm deep, strongly gelatinised subpellis of loose and irregularly oriented but near trama more dense and horizontally oriented, intricate, 2.0–3.5(–4.0) μm wide hyphae. **Acid-resistant incrustations** absent. **Hyphal terminations near the pileus margin** composed of 1–3(–5) cells until they join a basal branched cells, thin-walled; terminal cells (5.0–)7.5–22.9–38.0(–82.0) × (3.0–)4.0–5.6–7.5(–13.0) μm, mainly subulate and apically obtuse or attenuated, occasionally ellipsoid or clavate, and apically obtuse; subterminal cells usually unbranched, shorter, ellipsoid or subglobose, (4.5–)5.5–9.7–13.5(–25.0) × (2.5–)3.5–6.2–8.0(–10) μm. **Hyphal terminations near the pileus centre** similar, terminal cells (5.0–)10.5–23.1–35.5(–59.0) × (2.5–)3.5–5.3–7.0(–10.5) μm. **Pileocystidia near the pileus margin** dispersed, one-celled, narrowly cylindrical or subcylindrical, not distinctly narrowing towards the bases, thin-walled, (21.0–)31.0–46.7–62.0(–107.0) × 2.5–3.3–4.0(–4.5) μm, apically mainly obtuse and with 2.0–6.0 μm long appendage, contents optically empty or only with few dispersed granulations or crystals near terminal part, almost negative in sulfovanillin. **Pileocystidia near the pileus centre** similar, more frequent, (18.5–)25.0–42.7–60.5(–87.0) × (2.0–)2.5–3.5–4.0(–5.0) μm, near apical part often moniliform. Cystidioid and oleipherous hyphae in subpellis and context not observed.

**Examined specimens:** Slovakia, Tríbeč Mts., E of the village Podhorany, temperate forest, on ground under *Quercus*, *Carpinus* and *Fagus*, coord. 48°22′28″N, 18°08′42″E, 7.9.2022, M. Adamčík jun., SAV F-21192 (S948); Tríbeč Mts., Jelenec, N of the autocamp, temperate forest, on ground under *Quercus* and *Carpinus*, coord. 48°24′32″N, 18°12′8″E, 12.7.2010, S. Adamčík, SAV F-3178 (S886); Štiavnické vrchy Mts., Sebechleby, Stará hora, temperate forest, on ground under *Quercus* and *Carpinus*, coord. 48°16′54″N, 18°54′13″E, 13.7.2010, S. Adamčík, SAV F-4207 (S887); Krupinská planina Mts., Plášťovce, NE of Krašoria hill, temperate forest, on ground under *Quercus* and *Carpinus*, coord. 48°11′55.46″N, 19° 3′17.48″E, 4.7.2011, T. Christiansen, SAV F-3326 (S888).

**Note**: The *Virescentinae* were for a long period defined as a group without pileocystidia and with exclusively areolate pileus cuticle [65]. Some morphological classification concepts either expanded the concept of *Virescentinae* to group all *Russula* species of subgenus *Heterophyllidiae* without pileocystidia [11] or raised the status of areolate species to section rank [12]. Pileocystidia in *R. virescens* were first time observed by Buyck [66]. Pileocystidia of this species were overlooked for a long time, because they are very dispersed and inconspicuous, and almost absent near the pileus margin. Here we confirmed the presence of pileocystidia in the type species of subsection *Virescentinae*, *R. virescens*. Buyck and Adamčík [20] described the presence of pileocystidia also in other *Virescentinae* members with areolate pileus in North America.

## Discussion

The green-cracking Russulas *R. orientalovirescens*, *R. parvovirescens* *R. virescens* and *R. viridirubrolimbata* form a monophyletic group characterized morphologically by their distinctly areolate pileus margin combined with dominant green, grey and yellow colours. The only species in this group that also exhibits red colours near pileus margin is *R. viridirubrolimbata*, which seems to be easily recognizable from the other three species [64]. *Russula parvovirescens* is a North American species recognized by smaller basidiomata and more inflated (15–25 μm) subterminal cells in the pileipellis compared to *R. virescens* and *R. orientalovirescens* (Buyck et al. 2006). Unlike the older European (Romagnesi 1967, Sarnari 1998) and Chinese literature (Deng 2020), we confirmed that both the new Asian species and the European *R. virescens* have pileocystidia.

Our phylogenetic analysis of the ITS region confirmed that *R. orientalovirescens* was previously reported as *R. virescens* in Southeast Asia. This Asian species reported as *R. virescens* was represented by sequences from 42 collections of in Cao et al. (2013) from China, another two collections from China (Wang et al. 2022), four collections from Laos (Łuczaj et al. 2021) and one collection from Myanmar (Hosaka et al. 2021). Deng et al. (2020) reported eight sequences of the newly described species from China as *R*. aff. *virescens*, recognizing the difference between this Asian taxon and the European *R. virescens*, with the former showing shorter hyphal extremities in pileipellis. Three publicly available sequences originating from Thailand and Sri Lanka, and identified as *R. virescens*, grouped in *R. virescens* species clade and one Thai collection identified as the same species is placed in *R. parvovirescens* clade. We were unable to access Thai material or DNA corresponding to these sequences, but we doubt that both North American and European species occur in subtropical or tropical areas of Southeast Asia. It is possible, that publicly available sequences were used as template to edit low quality sequences, or there are more species with very little difference in ITS. The sequence KY649467, originating from Sri Lanka, was published as *R. virescens* and clustered in our ITS tree with European samples of this species, but it looks very different from the European species on the photograph: its pileipellis is not areolate, basidiomata are thin-fleshed and smaller, pileus is yellow-green and has bright orange spots near the pileus centre [38]. This indicates that Asian green-cracking Russulas require more detailed study using multilocus phylogenetic analysis and detailed morphological observations to distinguish them from their relatives.

Similar but according to ITS tree unrelated to the new species described here is *R. pseudocrustosa* G.J. Li & C.Y. Deng which also has areolate green pileus surface, but differs by having yellowish brown to reddish brown pileus colours, cream spore deposit and lower warts (0.2–0.5 μm high) on spores [5]. *Russula sribuabanensis* has also pruinose green pileus

but without distinct areolae, it differs also by low reticulate spore ornamentation, larger hymenial cystidia (61–109 × 8–14 µm) and rare or absent pileocystidia [31].

*Russula* species often receive good reputation for their culinary value due to their texture and flavour after cooking. In southern China and Southeast Asia, wild edible mushrooms are commonly gathered from the forests during rainy season (June to October) and sold in open air markets close to the collecting sites. Green-cracking Russulas are well-known as one of most valuable edible mushrooms and are widely consumed in southern China, Laos and Thailand. They are commonly known as Hed Lom Kra Khiao in Thai, Koh Ket Khiew or Khai Thao in Laos and Qingtoujun in Chinese [2,8,25,67,68]. In Laos, at least two forms were recorded; *R.* cf. *virescens* and *R.* cf. *viridirubrolimbata* [68], the first one probably corresponding to *R. orientalovirescens*. In Thailand, green-cracking Russulas are named as *R. virescens* and often are highly priced in northern provinces [2,67]. Local people often cook the mushroom by boiling, steaming or grilling and mix them with chili paste or use for folk remedies related to cancer prevention. In Yunnan province, southwestern China, green-cracking Russulas are among the most popular wild edible mushrooms [34]. Another Chinese ethnomyco-logical study reported that fresh and dry *R. virescens* are sold by ethnic groups in southwestern Yunnan. The mushroom was frequently stir-fried with garlic and fresh chili or cooked with meat soup [35]. In addition, the mushroom has been reported to contain carbohydrates, proteins, minerals and have a low glycemic index (GI) which is suitable for people who wish to control their blood sugar [2,69]. Another recent study reported that the Chinese *"R. virescens"* samples possess polysaccharides that have potential for inhibiting cancer cell proliferation and activating immune response [70]. Because of its culinary and medicinal values, we believe that the detailed description of our new species and comparisons with similar species can improve knowledge on the identification of this edible species and related species in the region. Our phylogenetic, morphological and ecological observations suggest that *R. orientalovirescens* is the most common popular edible fungus occurring in various subtropical deciduous and coniferous forests with dry season and distributed from Malay Peninsula, south China, Korean Peninsula to Japan.

## Acknowledgments

We wish to thank the anonymous reviewers for their constructive comments.

## Author contributions

**Conceptualization:** Komsit Wisitrassameewong, Slavomír Adamčík, Olivier Raspé.

**Data curation:** Komsit Wisitrassameewong, Slavomír Adamčík, Olivier Raspé.

**Formal analysis:** Komsit Wisitrassameewong, Olivier Raspé.

**Funding acquisition:** Komsit Wisitrassameewong, Slavomír Adamčík, Olivier Raspé.

**Investigation:** Komsit Wisitrassameewong, Slavomír Adamčík, Katarína Adamčíková, Song-Ming Tang, Narumon Tangthirasunun, Boontiya Chuankid, Olivier Raspé.

**Methodology:** Komsit Wisitrassameewong, Olivier Raspé.

**Resources:** Slavomír Adamčík, Olivier Raspé.

**Supervision:** Slavomír Adamčík, Olivier Raspé.

**Validation:** Komsit Wisitrassameewong, Olivier Raspé.

**Visualization:** Komsit Wisitrassameewong, Slavomír Adamčík, Katarína Adamčíková, Boontiya Chuankid.

**Writing – original draft:** Komsit Wisitrassameewong, Slavomír Adamčík.

**Writing – review & editing:** Komsit Wisitrassameewong, Slavomír Adamčík, Olivier Raspé.

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
