## [Decision Letter · Decision Letter 0]

29 Nov 2024

Dear Dr. Raspé,

We look forward to receiving your revised manuscript.

Kind regards,

Erika Kothe

Academic Editor

PLOS ONE

Journal Requirements: When submitting your revision, we need you to address these additional requirements. 1. Please ensure that your manuscript meets PLOS ONE's style requirements, including those for file naming. The PLOS ONE style templates can be found at https://journals.plos.org/plosone/s/file?id=wjVg/PLOSOne_formatting_sample_main_body.pdf and https://journals.plos.org/plosone/s/file?id=ba62/PLOSOne_formatting_sample_title_authors_affiliations.pdf 2. Please take this opportunity to be sure you have met all of our guidelines for new species. When publishing papers that describe a new fungal taxon name, PLOS aims to comply with the requirements of the International Code of Nomenclature for algae, fungi, and plants (ICN). The following guidelines for publication in an online-only journal have been agreed such that any scientific fungal name published by us is considered effectively published under the rules of the Code. Please note that these guidelines differ from those for zoological nomenclature.Effective January 2012, ""the description or diagnosis required for valid publication of the name of a new taxon"" can be in either Latin or English. This does not affect the requirements for scientific names, which are still to be Latin.Also effective January 2012, the electronic PDF represents a published work according to the ICN for algae, fungi, and plants. Therefore the new names contained in the electronic publication of a PLOS ONE article are effectively published under that Code from the electronic edition alone, so there is no longer any need to provide printed copies.For proper registration of the new taxon, we require two specific statements to be included in your manuscript.5. In the Results section, the globally unique identifier (GUID), currently in the form of a Life Science Identifier (LSID), should be listed under the new species name, for example:Hymenogaster huthii. Stielow et al. 2010, sp. nov. [urn:lsid:indexfungorum.org:names:518624]You will need to contact either Mycobank or Index Fungorum to obtain the GUID (LSID). 6. In the Methods section, include a sub-section called ""Nomenclature"" using the following wording (this example is for taxon names submitted to MycoBank; please substitute appropriately if you have submitted to Index Fungorum and use the prefix http://www.indexfungorum.org/Names/NamesRecord.asp?RecordID= ):The electronic version of this article in Portable Document Format (PDF) in a work with an ISSN or ISBN will represent a published work according to the International Code of Nomenclature for algae, fungi, and plants, and hence the new names contained in the electronic publication of a PLOS ONE article are effectively published under that Code from the electronic edition alone, so there is no longer any need to provide printed copies.In addition, new names contained in this work have been submitted to MycoBank from where they will be made available to the Global Names Index. The unique MycoBank number can be resolved and the associated information viewed through any standard web browser by appending the MycoBank number contained in this publication to the prefix http://www.mycobank.org/MB/. The online version of this work is archived and available from the following digital repositories: [INSERT NAMES OF DIGITAL REPOSITORIES WHERE ACCEPTED MANUSCRIPT WILL BE SUBMITTED (PubMed Central, LOCKSS etc)].All PLOS ONE articles are deposited in PubMed Central and LOCKSS. If your institute, or those of your co-authors, has its own repository, we recommend that you also deposit the published online article there and include the name in your article.A complete explanation of our guidelines for publishing new species can be found on our website: http://www.plosone.org/static/guidelines#fungal Special Cases – Algae, plant fossils, etc.Please take this opportunity to be sure you have met all of our guidelines for new species. For submissions describing new species that do not have formal registries, please include a sub-section called “Nomenclature” in the Methods section using the following wording:The electronic version of this article in Portable Document Format (PDF) in a work with an ISSN or ISBN will represent a published work according to the International Code of Nomenclature for algae, fungi, and plants, and hence the new names contained in the electronic publication of a PLOS ONE article are effectively published under that Code from the electronic edition alone, so there is no longer any need to provide printed copies.The online version of this work is archived and available from the following digital repositories: PubMed Central, LOCKSS [author to insert names of any additional repositories where the work will be deposited]. 3. In your Methods section, please provide additional information regarding the permits you obtained for the work. Please ensure you have included the full name of the authority that approved the field site access and, if no permits were required, a brief statement explaining why. 4. Thank you for stating the following financial disclosure: "O. Raspé: Mae Fah Luang University grant 641A01003, “Survey of edible fungi in dry dipterocarp forests of Chiang Mai Province, Thailand Komsit Wisitrassameewong: the Ecological Monitoring and Plant Specimen and Barcode References project P2250745, National Science and Technology Development Agency Slavomír Adamčík, Katarína Adamčíková: the Slovak Research and Development Agency projects APVV-15-0210 and APVV-20-0257." Please state what role the funders took in the study.  If the funders had no role, please state: ""The funders had no role in study design, data collection and analysis, decision to publish, or preparation of the manuscript."" If this statement is not correct you must amend it as needed. Please include this amended Role of Funder statement in your cover letter; we will change the online submission form on your behalf. 5. Thank you for stating the following in the Acknowledgments Section of your manuscript: "This work was financially supported by Mae Fah Luang University grant 641A01003, “Survey of edible fungi in dry dipterocarp forests of Chiang Mai Province, Thailand” to O. Raspé. Komsit Wisitrassameewong was supported by the Ecological Monitoring and Plant Specimen and Barcode References project P2250745, National Science and Technology Development Agency. The work of Slovak authors was supported by the Slovak Research and Development Agency projects APVV-15-0210 and APVV-20-0257." We note that you have provided funding information that is not currently declared in your Funding Statement. However, funding information should not appear in the Acknowledgments section or other areas of your manuscript. We will only publish funding information present in the Funding Statement section of the online submission form. Please remove any funding-related text from the manuscript and let us know how you would like to update your Funding Statement. Currently, your Funding Statement reads as follows: "O. Raspé: Mae Fah Luang University grant 641A01003, “Survey of edible fungi in dry dipterocarp forests of Chiang Mai Province, Thailand Komsit Wisitrassameewong: the Ecological Monitoring and Plant Specimen and Barcode References project P2250745, National Science and Technology Development Agency Slavomír Adamčík, Katarína Adamčíková: the Slovak Research and Development Agency projects APVV-15-0210 and APVV-20-0257." Please include your amended statements within your cover letter; we will change the online submission form on your behalf. 6. We note that your Data Availability Statement is currently as follows: All relevant data are within the manuscript and its Supporting Information files. Please confirm at this time whether or not your submission contains all raw data required to replicate the results of your study. Authors must share the “minimal data set” for their submission. PLOS defines the minimal data set to consist of the data required to replicate all study findings reported in the article, as well as related metadata and methods (https://journals.plos.org/plosone/s/data-availability#loc-minimal-data-set-definition). For example, authors should submit the following data: - The values behind the means, standard deviations and other measures reported;- The values used to build graphs;- The points extracted from images for analysis. Authors do not need to submit their entire data set if only a portion of the data was used in the reported study. If your submission does not contain these data, please either upload them as Supporting Information files or deposit them to a stable, public repository and provide us with the relevant URLs, DOIs, or accession numbers. For a list of recommended repositories, please see https://journals.plos.org/plosone/s/recommended-repositories. If there are ethical or legal restrictions on sharing a de-identified data set, please explain them in detail (e.g., data contain potentially sensitive information, data are owned by a third-party organization, etc.) and who has imposed them (e.g., an ethics committee). Please also provide contact information for a data access committee, ethics committee, or other institutional body to which data requests may be sent. If data are owned by a third party, please indicate how others may request data access. 7. When completing the data availability statement of the submission form, you indicated that you will make your data available on acceptance. We strongly recommend all authors decide on a data sharing plan before acceptance, as the process can be lengthy and hold up publication timelines. Please note that, though access restrictions are acceptable now, your entire data will need to be made freely accessible if your manuscript is accepted for publication. This policy applies to all data except where public deposition would breach compliance with the protocol approved by your research ethics board. If you are unable to adhere to our open data policy, please kindly revise your statement to explain your reasoning and we will seek the editor's input on an exemption. Please be assured that, once you have provided your new statement, the assessment of your exemption will not hold up the peer review process. 8. PLOS requires an ORCID iD for the corresponding author in Editorial Manager on papers submitted after December 6th, 2016. Please ensure that you have an ORCID iD and that it is validated in Editorial Manager. To do this, go to ‘Update my Information’ (in the upper left-hand corner of the main menu), and click on the Fetch/Validate link next to the ORCID field. This will take you to the ORCID site and allow you to create a new iD or authenticate a pre-existing iD in Editorial Manager. 9. Please amend either the title on the online submission form (via Edit Submission) or the title in the manuscript so that they are identical. 10. Please ensure that you refer to Figure 1 and 2 in your text as, if accepted, production will need this reference to link the reader to the figure. 11. We note you have included a table to which you do not refer in the text of your manuscript. Please ensure that you refer to Table 1 and 2 in your text; if accepted, production will need this reference to link the reader to the Table.

**Additional Editor Comments:**

The reviewers give important and helpfult comments that need to be addressed.

Reviewers' comments:

Reviewer's Responses to Questions

**Comments to the Author**

1. Is the manuscript technically sound, and do the data support the conclusions?

Reviewer #1: Yes

Reviewer #2: Yes

2. Has the statistical analysis been performed appropriately and rigorously?

Reviewer #1: Yes

Reviewer #2: Yes

3. Have the authors made all data underlying the findings in their manuscript fully available?

Reviewer #1: Yes

Reviewer #2: Yes

4. Is the manuscript presented in an intelligible fashion and written in standard English?

Reviewer #1: Yes

Reviewer #2: Yes

Reviewer #1: The manucript dealing with the diversity of green cracking Russulas and the formal description of a new commonly eaten species with major economical improtance is a valuable contribution that will in the future help to ensure food safety.

Please pay careful attention to unify formatting styles within the manuscript, especially regarding the two provided species descriptions.

An annotated manuscript file with specific comments is attached.

Reviewer #2: The paper by Wisitrassameewong et al. addresses the diversity of commercially valued Russula species within the Virescentinae lineage. Previous studies introduced considerable uncertainties by applying European species names to Asian collections. This study aims to clarify the taxonomy of Thai collections and includes the description of a novel species, Russula orientalovirescens, which is clearly distinct from the European Russula virescens.

At first glance, the study appears to be well-conducted, with adequate sampling and the application of standard phylogenetic methods. Notably, the authors included an analysis of environmental sequences, providing valuable insights into the distribution of the newly described R. orientalovirescens. The results are thoroughly discussed and compared with the current state-of-the-art literature.

However, several issues were raised during the review process:

Introduction and Hypotheses: The information provided in the Introduction does not clearly lead to the stated hypotheses. While the hypotheses and aims of the study are scientifically sound, they require clearer and more precise formulation.

Inconsistencies in Specimen Presentation: There is significant inconsistency in the documentation of the specimens analyzed for the newly described R. orientalovirescens. The authors stated that four specimens were analyzed, yet:

Only two specimens are listed in Table 1.

Three specimens are depicted in the phylogenetic trees.

Table 2 showcases the morphological variability of three specimens, including specimen OR1607, which is not listed in Table 1 or shown in the phylogenetic trees.

This raises concerns about whether the identity of OR1607 was molecularly confirmed. I strongly encourage the authors to include specimen OR1607 in an additional phylogenetic analysis.

Sequencing Effort: According to Table 1, only one specimen of R. orientalovirescens (OR1687) had multiple genetic regions sequenced. This contrasts with the more extensive sequencing efforts for European collections of R. mustelina and R. virescens. Similar efforts should be applied to R. orientalovirescens to ensure consistency and robustness of the analyses.

Terminology and Descriptions: Terminology and species descriptions should be more consistent throughout the manuscript (see detailed comments in the attached file).

References: The references require a thorough review for accuracy and completeness.

Additional minor comments and suggestions are included in the attached file.

**Do you want your identity to be public for this peer review?** For information about this choice, including consent withdrawal, please see our Privacy Policy

Reviewer #1: No

Reviewer #2: No

---

## [Author Response · Author response to Decision Letter 1]

6 Mar 2025

Manuscript PONE-D-24-43558

Response to the reviewers’ comments

Reviewer #1: The manuscript dealing with the diversity of green cracking Russulas and the formal description of a new commonly eaten species with major economical importance is a valuable contribution that will in the future help to ensure food safety.

Please pay careful attention to unify formatting styles within the manuscript, especially regarding the two provided species descriptions.

Reply: thank you for the positive feedback. We carefully revised the formatting styles.

Responses to each point raised by reviewer #1

Comment 1 (line 62): change to "Heterophyllidiae"

Reply: revised

Comment 2 (line 153): add space

Reply: added

Comment 3 (line 224): Please review the authors abbreviation, which has been unconsistently used in past publications and make a registry with the correct abbreviation in the International Plant Names Index to allow an unambiguous identification of the author.

Wissitrassameewong, K. Wisitrassameewong, Wissitr. or Wisitr.

Reply: revised. We used Wisitr., which is the form registered in Index Fungorum and IPNI.

Comment 4 (line 237): delete "in"

Reply: revised

Comment 5 (line 240): please add the colour codes of the chart that is cited in material and methods section similar to the following species description

Reply: the colour chart was not used to indicate the colour of the new species. This is inconsistent among both descriptions. We decided to indicate the colours without the colour codes

Comment 6 (line 244): taste should be bold, since it is not a subcharacter of odour.

Reply: revised

Comment 7 (line 244): 1A? 1B?

Reply: revised

Comment 8 (line 245): "×" is missing

Reply: revised

Comment 9 (line 282): genera names should be italic

Reply: revised

Comment 10 (line 284): add "to"

Reply: added

Comment 11 (line 291): change to "compared"

Reply: revised

Comment 12 (line 315): bold

Reply: revised

Comment 13 (line 316): italic

Reply: revised

Comment 14 (line 316): add taste

Reply: added

Comment 15 (line 331): it is bold in the previous description.

Reply: revised

Comment 16 (line 332): later in the text different spelling "thin-walled"

Reply: revised

Comment 17 (line 338): not bold in previous description.

Reply: revised

Comment 18 (line 351): The formatting style should be adapted to the format of the previous description.

Reply: revised

Comment 19 (line 355): genera names in italic

Reply: revised

Comment 20 (line 361): "Heterophyllidiae"

Reply: revised

Comment 21 (line 373): change to "exhibits"

Reply: revised

Comment 22 (line 376): change to "compared"

Reply: revised

Comment 23 (line 395): change to "indicates"

Reply: revised

Comment 24 (Table 2): please add information that all values are in µm.

Reply: added

Comment 25 (Figure 4): change to plural "Basidiospores"

Reply: revised

Comment 26 (Figure 4): change to plural "Marginal cells"

Reply: revised

Comment 27 (Figure 4): change to plural "scale bars"

Reply: revised

Comment 28 (Figure 5): change to plural "hyphal terminations"

Reply: revised

Comment 29 (Figure 7): see comments Fig. 4

Reply: revised

Comment 30 (Figure 8): see comments Fig. 5

Reply: revised

Comment 31 (Figure 1 ITS tree): replace "F. sylvatica" by "Fagus sylvatica" to unify the style.

Reply: revised

Comment 32 (Figure 1 ITS tree): replace "Slovania" by "Slovenia"

Reply: revised

Comment 33 (Figure 1 ITS tree): replace "Columbia" by "Colombia"

Reply: revised

Reviewer #2: The paper by Wisitrassameewong et al. addresses the diversity of commercially valued Russula species within the Virescentinae lineage. Previous studies introduced considerable uncertainties by applying European species names to Asian collections. This study aims to clarify the taxonomy of Thai collections and includes the description of a novel species, Russula orientalovirescens, which is clearly distinct from the European Russula virescens.

At first glance, the study appears to be well-conducted, with adequate sampling and the application of standard phylogenetic methods. Notably, the authors included an analysis of environmental sequences, providing valuable insights into the distribution of the newly described R. orientalovirescens. The results are thoroughly discussed and compared with the current state-of-the-art literature.

However, several issues were raised during the review process:

Introduction and Hypotheses: The information provided in the Introduction does not clearly lead to the stated hypotheses. While the hypotheses and aims of the study are scientifically sound, they require clearer and more precise formulation.

Reply: We propose this change (Lines 97-104):

Based on the molecular and morphological differences previously reported between the Asian green-areolate collections referred to as Russula virescens or R. cf. virescens in the literature on the one hand, and other green Virescentinae species and the European R. virescens on the other hand, we hypothesized that there is at least one additional Asian species of green-cracking Russula. Our aim was to confirm if there is phylogenetic signal (using both ITS and multi-locus data) to distinguish our recent Thai collections from European R. virescens and other described similar Asian Virescentinae members, and also to identify morphological differences between them in case they represent a distinct species.

Inconsistencies in Specimen Presentation: There is significant inconsistency in the documentation of the specimens analyzed for the newly described R. orientalovirescens. The authors stated that four specimens were analyzed, yet:

Only two specimens are listed in Table 1.

Three specimens are depicted in the phylogenetic trees.

Table 2 showcases the morphological variability of three specimens, including specimen OR1607, which is not listed in Table 1 or shown in the phylogenetic trees.

This raises concerns about whether the identity of OR1607 was molecularly confirmed. I strongly encourage the authors to include specimen OR1607 in an additional phylogenetic analysis.

Reply: we checked the number of samples in the table 1 and phylogenetic tree carefully. We added the information of OR1607 and OR1619 in the table 1.

Sequencing Effort: According to Table 1, only one specimen of R. orientalovirescens (OR1687) had multiple genetic regions sequenced. This contrasts with the more extensive sequencing efforts for European collections of R. mustelina and R. virescens. Similar efforts should be applied to R. orientalovirescens to ensure consistency and robustness of the analyses.

Reply: we tried and sequence the ITS, rpb2 and tef1 sequences of OR1607. We could only obtain the ITS sequence.

Terminology and Descriptions: Terminology and species descriptions should be more consistent throughout the manuscript (see detailed comments in the attached file).

Reply: revised accordingly

References: The references require a thorough review for accuracy and completeness.

Reply: revised accordingly

Additional minor comments and suggestions are included in the attached file.

Responses to each point raised by reviewer #2

Comment 1 (line 91-92): Hypothesis does not sound right to me. Since Cao et al. 2013 published their study, multiple species within Virescentinae were described. Additionally, there is Russula viridirubrolimbata which could potentially represent Cao´s "asian virescens". It would be beneficial better elaborate, on which basis you have assumed, that there could be higher diversity.

Reply: We rephrased our hypothesis and rationale for it. As explained above in the introduction, in all the Asian green Virescentinae, which do not include R. viridirubrolimbata because the pileus margin of the latter has red colours, the cuticle is cracking at most near the margin. The collections with green and almost entirely areolate cuticle are still referred to as R. virescens or R. cf. virescens despite dissimilarities in ITS sequences.

Comment 2 (line 92-95): This aim does not makes a sense to me. Previous phylogenetic studies clearly distinguished asian "virescentinae" from european. You have noted also this fact in the text above, when you state: "Soon it became apparent that at least part of so-called R. virescens in areas distant from Europe represent different species".

Therefore, I do not understand, why is important to distinguish your collections from european Russula virescens rather than closely related Russula viridirubrolimbata

Reply: see reply to comment 1

Comment 3 (line 190): In the presented tree, there is 42 samples

Reply: revised

Comment 4 (line235): you should unify units across the manuscript. Description of R. virescens is given in mm instead of cm

Reply: revised

Comment 5 (line 237): when mature instead of "in when"

Reply: revised

Comment 6 (line 238): areolae in centre?

Reply: revised

Comment 7 (line 245): 7.4–8.1–8.8(–9.2) × 6.0–6.4–6.9(–7.2)

Reply: revised

Comment 8 (line 246-250): about ornamentation are arranged somehow chaotically. I recommend to use same order as for R. virescens

Reply: revised

Comment 9 (line 248): isolated

Reply: revised

Comment 10 (line 275-276): information about cystidioid or oleipherous hyphae is absent

Reply: added

Comment 11 (line 418): superfluous

Reply: we prefer to keep it

Comment 12 (table 2): suggest to add also comparisons with the other species - R. parvovirescens and R. viridirubrolimbata, from literature

Reply: We added data on R. parvovirescens. However, we think that adding R. viridirubrolimbata would not be useful since this species can easily be differentiated from R. virescens based on macromorphology (red colours near pileus margin)

Comment 13 (table 2): In the description, you are referring to pleurocystidia as hymenial cystidia on lamellae sided / cheilocystidia on lamellae edges

Reply: revised

---

## [Decision Letter · Decision Letter 1]

24 Mar 2025

Russula orientalovirescens sp. nov., a common Southeast Asian edible fungus is different from the European look-alike R. virescens

PONE-D-24-43558R1

Dear Dr. Raspé,

We’re pleased to inform you that your manuscript has been judged scientifically suitable for publication and will be formally accepted for publication once it meets all outstanding technical requirements.

Kind regards,

Erika Kothe

Academic Editor

PLOS ONE

Reviewers' comments:

Reviewer's Responses to Questions

**Comments to the Author**

Reviewer #1: All comments have been addressed

Reviewer #2: All comments have been addressed

2. Is the manuscript technically sound, and do the data support the conclusions?

Reviewer #1: Yes

Reviewer #2: Yes

3. Has the statistical analysis been performed appropriately and rigorously?

Reviewer #1: Yes

Reviewer #2: Yes

4. Have the authors made all data underlying the findings in their manuscript fully available?

Reviewer #1: Yes

Reviewer #2: Yes

5. Is the manuscript presented in an intelligible fashion and written in standard English?

Reviewer #1: Yes

Reviewer #2: Yes

Reviewer #1: Dear Authors, thank you for addressing the proposed changes. I consider this research a valuable contribution to the knowledge on the diversity of the genus Russula in Thailand.

Reviewer #2: All my comments were properly addressed and manuscript was improved accordingly. I still recommend thorough review of the manuscript for small typos, but other than that I do not have any other comments

**Do you want your identity to be public for this peer review?** For information about this choice, including consent withdrawal, please see our Privacy Policy

Reviewer #1: No

Reviewer #2: No

---

## [Editor Report · Acceptance letter]

PONE-D-24-43558R1

PLOS ONE

Dear Dr. Raspé,

I'm pleased to inform you that your manuscript has been deemed suitable for publication in PLOS ONE. Congratulations! Your manuscript is now being handed over to our production team.

Kind regards,

on behalf of

Prof. Dr. Erika Kothe

Academic Editor

PLOS ONE